# In Vitro and In Vivo Models for Studying SARS-CoV-2, the Etiological Agent Responsible for COVID-19 Pandemic

**DOI:** 10.3390/v13030379

**Published:** 2021-02-27

**Authors:** Rafael B. Rosa, Willyenne M. Dantas, Jessica C. F. do Nascimento, Murilo V. da Silva, Ronaldo N. de Oliveira, Lindomar J. Pena

**Affiliations:** 1Department of Virology, Aggeu Magalhães Institute (IAM), Oswaldo Cruz Foundation (Fiocruz), Recife 50740-465, Brazil; rafael.rosa@cpqam.fiocruz.br (R.B.R.); jessica.frutuoso7@gmail.com (J.C.F.d.N.); 2Rodents Animal Facilities Complex, Federal University of Uberlandia, Uberlandia 38400-902, Brazil; murilo.vieira@ufu.br; 3Department of Chemistry, Federal Rural University of Pernambuco (UFRPE), Recife 52171-900, Brazil; willyenne.dantas@ufrpe.br (W.M.D.); ronaldo.noliveira@ufrpe.br (R.N.d.O.)

**Keywords:** SARS-CoV-2, COVID-19, in vivo models, in vitro models

## Abstract

The emergence and rapid worldwide spread of severe acute respiratory syndrome coronavirus 2 (SARS-CoV-2) has prompted the scientific community to rapidly develop in vitro and in vivo models that could be applied in COVID-19 research. In vitro models include two-dimensional (2D) cultures of immortalized cell lines or primary cells and three-dimensional (3D) cultures derived from lung, alveoli, bronchi, and other organs. Although cell-based systems are economic and allow strict control of experimental variables, they do not always resemble physiological conditions. Thus, several in vivo models are being developed, including different strains of mice, hamsters, ferrets, dogs, cats, and non-human primates. In this review, we summarize the main models of SARS-CoV-2 infection developed so far and discuss their advantages, drawbacks and main uses.

## 1. Introduction

In December 2019, cases of severe pneumonia of unknown origin were reported in Wuhan, China [1]. After phylogenetic analysis, the etiological agent of the severe acute respiratory syndrome was identified as a novel virus member of the *Coronaviridae* family [2]. The virus was named severe acute respiratory syndrome coronavirus 2 (SARS-CoV-2) by the International Committee on Taxonomy of Viruses (ICTV) [3]. The World Health Organization (WHO) referred the disease caused by the new coronavirus as “COVID-19” in February 2020 [4]. Rapidly, COVID-19 took global proportions and after three months following its emergence, WHO declared it a pandemic [5]. 

COVID-19 was initially thought of as primarily a respiratory disease, but soon it was recognized that SARS-CoV-2 could affect many body systems, including the respiratory, gastrointestinal, hepatic, ocular, cardiovascular, and neurological [6,7,8]. Thus, a considerable number of research models have been developed to mimic the disease pathophysiology under experimental conditions. 

Several cellular and animal models have been used for studying SARS-CoV-2 infection. In vitro models are useful for studying virus biology under highly controlled conditions, but they often fail to recapitulate the complexity of human body systems [9,10]. In vivo models for the SARS-CoV-2 infection together with other methodologies have the potential to elucidate the natural history of the disease, contributing to the discovery of antivirals and vaccines against COVID-19 [11]. However, in vivo studies are costly, require BSL-3 animal facilities, and raise ethical concerns. 

In this review, we summarize the in vitro (2D and 3D cell culture) and in vivo models that have been developed for studying SARS-CoV-2 biology and discuss their advantages, drawbacks, and main uses.

## 2. In Vitro Models

In vitro models are based on two-dimension (2D) or three-dimension (3D) cultures of primary or immortalized cells and tissues. Each methodology has its own applicability and disadvantages for SARS-CoV-2 studies as described in the following sections and summarized in Table 1. In vitro models comply with the ethical desire for reducing the use of animal models and can answer relevant questions regarding SARS-CoV-2 biology and treatment. They are inexpensive, fast, and allow for the study of specific cellular targets, which could not be assessed in a macroscopic system. However, cell models do not resemble the complexity of a whole organism and translatability of in vitro-generated data to in vivo models can be particularly challenging [12].

### 2.1. 2D-Cell Models: Immortalized Cells

Monolayer culture of immortalized cells has been used widely for isolating the new coronavirus. The Vero cell line, which is derived from African green monkey kidney, is the most largely used cell line for vaccine production and has been one of the most common cell lines for SARS-CoV-2 isolation around the world. Vero cells are known to express high levels of angiotensin-converting enzyme 2 (ACE2) [13], the cellular receptor used for entry by SARS-CoV-2 [14].

Vero E6 and Huh-7 cells (human hepatocarcinoma) were used to isolate the virus from the first Wuhan patients diagnosed with COVID-19. The presence of cytopathic effect occurred after six days post infection (dpi) [15]. SARS-CoV-2 was also isolated from the first reported South Korean COVID-19 patient using Vero cells. Once again, no cytopathic effect was observed until 5 dpi. However, three days after a blind passage, the cytopathic effect characterized by rounding and detachment of cells was seen. Viral particles were visualized by transmission electron microscopy, RNA was sequenced, and the phylogenetic analysis showed the similarity of the strains previously isolated [16]. Harcourt and colleagues isolated the virus from oropharyngeal (OF) and nasopharyngeal (NF) samples in Vero CCL-81 cells and observed the cytopathic effect 2 dpi. Virus was detected using real-time quantitative polymerase chain reaction (qRT-PCR) analysis and confirmed by sequencing. Genome sequences of virus cultivated in Vero E6 were also obtained and showed similarity with the parental strain. SARS-CoV-2 grows to similar titers in both Vero-based infection models (CCL-81 and E6), but plaque formation was more visible in Vero E6 cells [17]. Araujo et al. isolated SARS-CoV-2 from the first reported COVID-19 patients in Brazil in Vero-E6 cells and then compared the replication and viral cytopathic effect (CPE) in three Vero cell lines (E6, CCL-81 and hSLAM). The virus grew to similar titers in these cells, but CPE was more pronounced in CCL-81 cells compared to the others. Plaque formation of the Brazilian strain was not clear in Vero E6 cells, but could be seen in Vero CCL81 cells, indicating that different SARS-CoV-2 strains may produce distinct cytopathology in continuous cell lines [18]. SARS-CoV-2 infectivity was studied in human airway epithelium (HAE) and Vero E6. The virus grew to similar titers in both cells, and infection could be blocked by convalescent patient serum. The virus was released apically, not from basolateral side, in HAE cells [19]. Studies with different virus after several passages in Vero cells reported the introduction of adaptive and deleterious mutations in the viral Spike protein that altered SARS-CoV-2 infectivity and its phenotypic characteristics [20,21]. The genetic changes observed during repeated viral passages demonstrate the ability of the virus to adapt quickly to the host. This highlights the need to carefully keep track of viral passage number and in-depth genetic characterization of virus strains prior to perform in vitro and in vivo studies.

Caco-2 cells, a line of colorectal adenocarcinoma, were used for isolating SARS-CoV-2 from infected patients traveling from the city of Wuhan to Frankfurt, Germany [22]. The infection capacity and production of cytokines and chemokines were tested in Caco-2 cells infected with SARS-CoV-1, SARS-CoV-2, MERS-CoV, H1N1pdm, or H5N1 viruses. Caco-2 was susceptible to all tested viruses. SARS-CoV-2 infection in this cell line resulted in low or no pro-inflammatory response [10]. Aiming to establish a permissive cell model with Caco-2 cells to detect possible therapeutic targets, Bojkova and colleagues inoculated the previously isolated strain by Hoehl and observed the cytopathic effect within 24 h. Proteomic analysis showed that Caco-2 cells could change their metabolism during viral replication [23].

A549 cells, a human alveolar basal epithelial carcinoma cell line, expressing ACE2 were used to identify host factors required for SARS-CoV-2 replication. Cellular genes were deleted using the clustered regularly interspaced short palindromic repeats (CRISPR) technique and then cells were checked for infection. This methodology allowed assessing the impact of each protein-coding gene in the human genome on viral replication and may pave the way to in vivo pathogenesis studies and to discover countermeasures [24].

Immortalized cells cultured in monolayers have been essential for detecting potential anti-SARS-CoV-2 drugs. Given the SARS-CoV-2 replicative efficiency in Vero E6 cells, they were used for testing drugs and antivirals listed in TargetMol library. After screening, three candidates reduced CPE when compared to untreated cells within 48 h. Cepharanthine was highlighted, since this drug previously showed an inhibitory effect on other coronaviruses (SARS-CoV and HCoV-OC43). After a time-of-addition experiment, the authors proposed that cepharanthine acted since the virus entry until post-viral infection [25]. Using Vero E6 cells as well, the compounds Boceprevir, GC-376 and calpain II/XII inhibitor were tested for SARS-CoV-2 inhibition through evaluation of the cytopathic effect and determination of EC_50_ in concentrations ranging from 0.49 to 3.37 μM. The proposed mechanism of action of these drugs was protease inhibition [26]. The compound 6-thioguanine (6-TG), a potent cytotoxic agent used in the treatment of acute leukemia, was tested to inhibit SARS-CoV-2 due to viral papain-like protease inhibition in Vero E6 and Calu-3 cells. Up to concentrations of 50 µM, the compound showed no cytotoxicity for both cells. The drug was able to inhibit virus replication with EC_50_ of 0.647 ± 0.374 μM for Vero and EC_50_ of 0.061 ± 0.049 μM for Calu-3 [27].

Remdesivir (RDV), a nucleoside analog initially planned for the treatment against Ebola and Marburg viruses, was another drug tested for anti-SARS-CoV-2 activity in 2D models. Prior to antiviral analysis, authors verified the reproduction of SARS-CoV-2 in Vero CCL-81, Vero E6, Huh-7 and Calu-3 2B4. Huh-7 showed low permissiveness to SARS-CoV-2 in comparison to the other cell lines. Only Vero E6 and Calu-3 cells were used in the antiviral tests with the RDV and GS-441524 (RDV prodrug). In Vero E6 cells, RDV (EC_50_ = 1.65 μM) proved to be less potent than GS-441524 (EC_50_ = 0.47 μM) and in Calu-3 cells the opposite was observed (RDV EC_50_ = 0.28 μM; GS-441524 EC_50_ = 0.62 μM) [28]. Other FDA-approved drugs have been tested against SARS-CoV-2 in Vero cells, including chloroquine/hydroxychloroquine and ivermectin. Both drugs reduced SARS-CoV-2 replication in Vero cells [29,30]. This effect was observed for chloroquine and hydroxychloroquine when administered pre and post-infection and the latter was found to be more potent than the former [30]. Ivermectin treatment of infected Vero cells can cause a ~5000-fold reduction in virus RNA levels [29]. However, some points need to be taken into consideration regarding the use of these drugs and their respective studies. These limitations include ivermectin’s low water solubility, lack of comparative standard drugs in antiviral tests, absent evaluation of treatment efficacy according to disease severity, and the possible combination with other drugs to maximize antiviral efficacy. Dose adjustment and evaluation of other administration routes are also important points that need to be further assessed. Several clinical trials have been concluded and others are ongoing and the clinical benefits of using these compounds in COVID-19 patients remains uncertain [31,32].

### 2.2. 3D-Cell Models: Explants and Organoids

The use of primary cells from human airway epithelium (HAE) for isolation and cultivation of other coronaviruses has already been described in previous studies [33]. HAE cells are isolated from human lung donors or individuals undergoing lung transplantation and resemble the human bronchial environment, one of the targets for SARS-CoV-2 infection. Upon isolation, these cells are cultured and differentiated on porous supports at an air–liquid interface (ALI), in which the apical side is in contact with air. HAE cells represent a complex tissue with differentiated ciliated, goblet and basal cells and retain the highest similarity to human airway epithelium physiological conditions found in vivo [34,35].

HAE cells were used to isolate and discover SARS-CoV-2 for the first time upon its emergence in Wuhan, China. Bronchoalveolar lavage fluids from three patients with pneumonia of unknown cause were inoculated in a primary culture of lung epithelial cells expanded in an ALI system at 37 °C and 5% carbon dioxide (CO_2_). The presence of the virus was visualized by transmission electron microscopy and confirmed by RT-PCR targeting a region of the viral RNA-dependent RNA-polymerase (RdRp) gene of pan β-CoV. Infected HAE showed cytopathic effects 96 h after inoculation, which was characterized by a lack of ciliary movement [15].

The ex vivo replication and tropism of SARS-CoV-2 has been studied in HAE cells. Hou and colleagues used HAE obtained from different zones of the respiratory tract (nasal, bronchial, bronchiolar, and alveolar tissue) and air absorption capacity to evaluate the infection with a SARS-CoV-2 reporter virus. Following the expression analysis of the angiotensin II receptor (ACE2) and the viral titers obtained from each cell, HAE from proximal airway (nasal cavity and large bronchi and alveoli) were found to be more susceptible to infection than distal regions. Nasal submucosal glands, type I and II pneumocytes, microvascular endothelial cells, fibroblasts and UNCNN2TS immortalized nasal cells were also tested but showed low viral titers or no infection signs [36].

Stem cells were induced to produce type 2 alveolar epithelial cells (iAT2) in an ALI system aiming to identify gene responses to SARS-CoV-2 infection. Since ACE2 and TMRPSS2 are required for SARS-CoV-2 infection in human cells [37], their gene expression profile was measured in different induced lung cell types. ACE2 and TMPRPSS2 expression was identified in differentiated basal, secretory and ciliated cells, suggesting that they are susceptible to the virus [38,39].

Because the most severe forms and mortality of COVID-19 are seen more often in men than women, the effects of estrogen on the replication of SARS-CoV-2 have been studied under controlled conditions in normal human bronchial epithelial cells (NHBE). According to a recent the study, estrogen reduces the expression of ACE2 and this could modulate the severity of COVID-19 in women [40]. NHBE cells were also used for identifying the expression of several genes in response to SARS-CoV-2 infection, showing an induction of chemokines, cytokines, and interleukins, with limited IFN-I and III responses [41].

Primary cells have also been used in SARS-CoV-2 antiviral research. The broad-spectrum oral antiviral β-D-N4-hydroxycytidine (NHC) was tested on infected human tracheobronchial epithelial cells cultured in an ALI system for 6 to 8 weeks. NHC decreased up to 3 logs the viral replication in concentrations between 0.1 and 10 µM without cytotoxic effects, which indicates its potential utility as an antiviral for treating SARS-CoV-2 infection [42]. Camostat and Nefamostat, both protease inhibitors, were tested on human nasal epithelial cells (HNE) infected with SARS-CoV-2, in concentrations between 0.001 and 1 g/mL, and a decrease in viral replication was achieved. RDV has also been tested on SARS-CoV-2 infected primary cells. Among the tests that have been carried out with this compound, lung epithelial cells cultured in an ALI system were used and showed good antiviral results with cytotoxic concentrations up to 10 µM. The percentage of viral inhibition reached 100% at the highest concentrations used and EC_50_ ranged between 0.001 and 0.009 µM [28].

Cellular models based on HAE offer similarity to physiological conditions and maintain many of the important markers and functions from the donor tissue [35]. Despite the great advantages of using primary cells, mainly regarding more realistic virus-host interactions, models based on immortalized cell monolayers are easy to use, less time-consuming and bypass ethical concerns associated with the use of animal and human tissue [34].

The use of 3D-cell models in SARS-CoV-2 research has gained attention due to their similarity with the host organism. Organ-derived explants (ex vivo models) or organoids produced by induced stem cells are beginning to be widely applied in the study of respiratory viral infections [43]. Results obtained from tissues are important for mapping their possible response to viral infection and finding new therapeutic targets. Patient-derived explants have the advantage of preserving overall tissue architecture and complexity when compared to organoids, but the latter can be more easily standardized compared to the former.

COVID-19 patients present symptoms in several systems, including the respiratory, gastrointestinal, hepatic, ocular, and cardiovascular [6,7]. Investigation of possible SARS-CoV-2 permissive cells in 13 human tissues was performed through the evaluation of ACE2 expression. Pulmonary AT2 cells, liver cholangiocytes, colonocytes, esophageal keratinocytes, ileum and rectum endothelial cells, stomach epithelial cells and renal proximal tubules presented the highest ACE2 levels, indicating that these cells could be susceptible to SARS-CoV-2 and be used in infection models [44]. With the same goal, Singh and colleagues used scRNA-seq to map different cell types and tissues permissive to the virus. Gastrointestinal and respiratory tract tissues were permissive to SARS-CoV-2, as well as spermatogonia and cells from the placenta [45]. The replicative potential of SARS-CoV-2 has been tested in lung, bronchial and connective tissues obtained from donor patients who underwent surgery. The virus was inoculated at a concentration of 5 × 10^5^ TCID_50_/mL at 37 °C (lung and bronchi) or 33 °C (conjunctive tissue) for 1 h. Samples were collected after 1, 24, 48, 72, and 96 h and viral load was measured by titration. All tissues proved to be permissive to infection with an increase of viral titers up to 2 logs [10]. Chu and colleagues analyzed SARS-CoV and SARS-CoV-2 infection in lung tissues donated by patients who underwent surgical procedures. SARS-CoV-2 was more infectious when compared to SARS-CoV. In the same study, immune and pro-inflammatory responses were also evaluated. SARS-CoV infection led to increased lung gene expression of IFN I, II, and III and 11 of 13 cytokines/chemokines tested. SARS-CoV-2 infection, on the other hand, did not alter the expression of IFN and induced the expression of only 5 cytokines/chemokines analyzed [46].

Given the hepatic impairment in patients with COVID-19, the permissiveness to SARS-CoV-2 infection was evaluated in hepatocyte and cholangiocyte organoids derived from human pluripotent stem cells (iPSCs). Both organoids supported productive SARS-CoV-2 replication and had similar chemokine responses as COVID-19 tissues from autopsy cases [47].

HPSCs-derived lung and colon organoids (hPSCs-LO and hPSCs-CO, respectively) were developed and used for evaluating the anti-SARS-CoV-2 activity of imatinib, mycophenolic acid (MPA) and quinacrine dihydrochloride (QNHC). Both organoids were permissive for SARS-CoV-2 infection. Similar to what is seen in COVID-19 patients, infected hPSC-LOs (particularly alveolar type-II-like cells) revealed upregulation of cytokines and chemokines [48].

Severe cases of COVID-19 can display neurological complications that are manifested by headache, confusion, seizure, and encephalopathy. Thus, the direct involvement of SARS-CoV-2 in the CNS has been studied in experimentally infected 3D neurospheres and brain organoids, which model the early characteristics of neurogenesis and the human cortical brain, respectively. Evidence provided independently using these 3D systems support autopsy findings that the human brain is permissive to SARS-CoV-2 infection and supports productive virus replication [49,50,51].

ACE2 expression was found to be low in human corneal and conjunctiva samples, which could explain the inferior susceptibility of these tissues to SARS-CoV-2 and the low frequency of conjunctivitis in COVID-19 patients [52]. Only 0 to 5.2% of PCR-positive COVID-19 patients have detectable SARS-CoV-2 RNA in conjunctival swabs or tears [53,54]. However, these findings have been contested by others. Makovoz et al. (2020) used eye organoids and adult human ocular cells and found that ocular cells express ACE2 and TMPRSS2, essential proteins for SARS-CoV-2 viral entry, and that these cells are susceptible to the virus. They found higher levels of viral replication in the limbus (border of the cornea and the sclera) compared to the central cornea area [55].

Epithelial cells of human renal proximal tubules (KPTEC) were cultured using a 2D conditional reprogramming system (CR) and 3D organoids cultures as physiological ex vivo kidney models. ACE2 expression in 2D CR culture was about half the one observed in 3D organoids culture. Pseudovirion assays with a SARS-CoV spike (S) protein construct demonstrated that CR KPTECs were permissive for SARS-CoV infection, suggesting that these models could be used to study SARS-CoV-2 nephropathology [9]. Susceptibility of 3D kidney organoids derived from human embryonic stem cells to SARS-CoV-2 infection has been directly confirmed [56].

Cumulative evidence from basic research and clinical studies has recognized COVID-19 as a multisystem disorder that also affects the vascular system. Monteil and co-workers engineered human capillary organoids from induced pluripotent stem cells (iPSCs) and demonstrated that human blood vessel organoids could be readily infected by SARS-CoV-2 [56].

## 3. Animal Models

An appropriate animal model develops the desired viral infection with clinical signs similar to those observed in humans. It is possible to elucidate virus biology and the mechanisms of infection using in vivo approaches, thus allowing the development of new drugs and vaccines. Therefore, the prevention and control of SARS-CoV-2 require the establishment of appropriate animal models for viral study [57]. Experimental studies have demonstrated that mice, hamster, rabbits, bats, cats, dogs, ferrets, and non-human primates can be infected by SARS-CoV-2 [58]. The following sections describe the most important animal models currently being used in SARS-CoV-2 research and their main characteristics are summarized in Table 2, Table 3 and Table 4.

### 3.1. Mice

ACE2 is the receptor used by SARS-CoV-2 to enter human cells [14]. SARS-CoV-2 presents a high affinity for human ACE2, but low affinity for ACE2 from other organisms [59]. Previous studies have shown that SARS-CoV-1 does not replicate efficiently in wild-type mice, thus requiring animal adaptation via serial viral passages or the development of transgenic mouse models capable of expressing human ACE2 [60,61,62,63,64]. This limitation of viral replication in wild-type mice can also be observed in animals infected with SARS-CoV-2, since there are no efficient interactions between the viral Spike protein (S) and the murine ACE2 [15]. Mice expressing human ACE2 (hACE2) were generated using CRISPR/Cas9 technology. Young or aged, wild-type or hACE2-producing C57BL/6 mice were infected intranasally with SARS-CoV-2. Infected hACE2 mice developed high viral loads in trachea, lung, and brain. Although no deaths were observed, aged hACE2 mice presented interstitial pneumonia and elevated cytokine levels. After intragastric virus inoculation, a productive infection was observed, which led to pathological pulmonary changes in these animals [65].

Previous studies with SARS-CoV-1 have developed a transgenic mouse called HFH4-hACE2, generated from C3B6 mice. This mouse expresses hACE2 under the control of HFH4/FOXJ1 promoter on lung ciliated epithelial cells [61,66]. This model expressed high levels of hACE2 in the lung, but different levels in other tissues, including brain, liver, kidney and gastrointestinal tract. Infected mice can lose more than 20% of body weight and die due to lethal encephalitis [61]. This model was recently applied in studies with SARS-CoV-2. Infected animals had typical interstitial pneumonia and clinical signs similar to those observed in COVID-19 patients. The virus was detected in the lungs, the main site of infection, although viral RNA has also been found in the eye, heart, and brain of some animals. Pre-exposure to SARS-CoV-2 was able to protect mice from re-infection [67].

hACE2 mice infected intranasally with SARS-CoV-2 showed up to 8% of weight loss until 5 dpi. Animals also showed bristled fur, lordosis position and decreased response to external stimuli. Viral loads were detected in the lungs at 1, 3, 5, and 7 dpi. Interestingly, viral RNA was detectable 1 dpi in the intestine but not in the following days. Although viral loads were detected in the intestine, no virus was isolated, suggesting that a residual inoculum reached the intestine by swallowing. Histopathological exams showed that animals developed interstitial pneumonia, presence of scattered dark reddish-purple areas in the lung, and palpable nodules in the lungs [68].

hACE2 interaction with human TMPRSS2 (hTMPRSS2) improves cellular entry of SARS-CoV [69]. Recently, it has been discovered that hTMPRSS2 is also important for priming of SARS-CoV-2 protein S and inhibition of this protease prevents viral cell entry [14]. New studies may be carried out considering the development of mouse models expressing both hTMPRSS2 and hACE2, which could create a new transgenic lineage capable of effectively recapitulating the disease [70].

Another approach to circumvent the absent affinity of SARS-CoV-2 for mouse ACE2 (mACE2) is using reverse genetics to remodel the protein S-mACE2 binding interface. This resulted in a recombinant virus (SARS-CoV-2 MA) capable of using mACE2 for entering cells. SARS-CoV-2 MA was able to replicate in the upper and lower airways of young and adult BALB/c wild-type mice. Aged mice showed a more pronounced disease [71]. SARS-CoV-2 MA infection resulted in high viral titers in lung tissue 2 dpi, but the virus was rapidly eliminated 4 dpi under the same conditions, parental SARS-CoV-2 virus was not infectious for the mice. Airway inflammation was observed in histopathological analysis performed 2 dpi, associated with high levels of viral antigen staining [72].

Pulmonary transduction of mice with viral vectors encoding human ACE2 has successfully rendered conventional mice susceptible to SARS-CoV-2 infection, clinical disease and lung pathology. This approach offers the advantage of using commercially available mouse strains from multiple backgrounds and genetic modifications. Viral vectors that have been used so far for transducing mouse lungs with hACE2 include replication-defective human adenovirus 5 (Ad5) [73], adeno-associated virus (AAV) [74], and Venezuelan equine encephalitis replicon particles (VEEV) [75].

Recently, Rathnasinghe et al. compared side by side the replication and morbidity of K18-hACE2 transgenic model to adenovirus vector-mediated delivery of hACE2 to the mouse lung. They demonstrated that hACE2 adenovirus-transduced mice infected with SARS-CoV-2 had no clear clinical signs of disease, and lower viral replication was limited to the nasal turbinates and lung. In contrast, K18-hACE2 mice developed a severe disease with high lethality manifested by weight loss, lethargy, ruffled fur, hunched posture, and labored breathing. These transgenic animals also had 2 to 3 logs higher levels of viral replication in the nasal turbinates, lung and brain compared to the transduced model [76].

Virus adaptation to non-human organisms via serial passages can also be used to develop a mice-infectious strain. SARS-CoV-2 was inoculated on BALB/c mice using the intranasal route and the virus was recovered from pulmonary homogenates. Viral RNA was detected at high levels from the third passage and remained high until the sixth passage. The strain obtained in the sixth passage was then inoculated in groups of young (6-weeks-old) and aged (6-months-old) mice. The adapted SARS-CoV-2 efficiently infected both young and aged mice resulting in moderate pneumonia and inflammatory responses. This animal model was used for testing the effectiveness of an anti-SARS-CoV-2 vaccine based on RBD subunit. The vaccine prototype triggered antibody production, showed potent neutralizing effects, and conferred total protection against the infection [77]. 

### 3.2. Golden Syrian Hamsters

The golden Syrian hamster (*Mesocricetus auratus)* is an animal model widely used in studies with SARS-CoV due to its permittivity to this viral infection [62]. The SARS-CoV-2-binding domain of ACE2 presents a high degree of similarity between hamsters’ and humans’ receptors, encouraging the use of golden Syrian hamsters in SARS-CoV-2 studies [78], despite the lack of certain reagents for immunological studies in this species. These animals presented viral replication in the lungs after SARS-CoV-2 intranasal inoculation. Histopathological analysis showed that the lungs developed marked pulmonary edema, inflammation, and cell death. Infected hamsters lost weight and showed an increased respiratory rate [79]. Immunohistochemistry evaluations of hamsters inoculated intranasally with SARS-CoV-2 demonstrated the presence of viral antigens in the nasal and bronchial mucosa, epithelial cells and in areas of pulmonary consolidation 2 and 5 dpi, followed by pneumocyte hyperplasia 7 dpi. Viral antigens were also observed in duodenum epithelial cells and viral RNA was detected in feces [80]. Moreover, Syrian hamsters developed a strong neutralizing antibody response against SARS-CoV-2, which provides immunity to subsequent virus rechallenge [81]. The role of STAT2 in SARS-CoV-2 infection has been demonstrated using a genetically engineered hamster model. STAT2-knockout hamsters have higher viral pulmonary titers, viremia, and systemic spread when compared to wild type animals. This indicates the importance of STAT2 in attenuate viral dissemination in the body. Therefore, STAT2-knockout hamsters had lower leukocyte infiltration, reduced pulmonary pathology, and absence of pneumonia [82].

The hamster model can mirror certain epidemiological features of COVID-19 in humans, such as the effect of age and sex in the severity of the disease. Thus, the age of experimental animals should be considered during the development of a hamster model for SARS-CoV-2 infection. Osterrieder et al. followed the course of SARS-CoV-2 infection in young and aged Syrian hamsters. Although viral replication in the upper and lower respiratory tract occurred regardless of animals’ age, hamsters infected at older ages experienced a more pronounced weight loss compared to younger animals. Histopathological analysis showed an important age-dependent influx of immune cells into the lungs, which happened earlier and stronger in young animals. Older hamsters developed conspicuous alveolar and perivascular edema, which indicates vascular leakage. Young animals had a rapid recovery from 14 dpi [83]. Studies using hamsters have also shown that infected animals transmitted the virus to healthy animals when co-housed. Although the infected animals exhibited a milder disease, hamsters can still be considered potential models for transmissibility studies [79].

Syrian hamsters were also used in the evaluation of transmission considering the use of masks. For this, the animals were placed in parallel cages in a closed system separated by a porous polyvinyl chloride air partition with unidirectional air flow. To assess the transmissibility of the virus, a surgical mask was placed between the cages. In the absence of the mask between the cages, the rate of transmission between animals was 66.7% while in the presence of the mask the rate of transmission decreased significantly to 16.7%. Histopathological changes, the amount of virus found in the respiratory tract and the expression of antigens in naive hamsters protected by the mask were significantly milder than in shredded hamsters [84].

Lastly, Syrian hamsters have proven to be a suitable to evaluate vaccines [85,86,87] and antiviral drugs [88,89,90] against SARS-CoV-2.

### 3.3. Ferrets

Ferrets (*Mustela putorius furo*) are a popular model for respiratory infections because their lung physiology and pathological response resembles that of humans. Unlike mice and rats, ferrets exhibit the cough reflex [91] and are widely used as a model for transmission and pathogenesis of respiratory viruses. Since coughing is the most frequently reported symptom in cases of SARS-CoV-2 infection, these animals represent good models for COVID-19 [92]. Similar to hamsters, ferrets are a well-suited animal model for testing medical countermeasures against SARS-CoV-2, including drugs [93] and vaccines [94]. SARS-CoV-2 intranasal inoculation in ferrets led to viral RNA detection in nasal lavages after 2, 4, 6, and 8 days. Viral RNA was also detected in some rectal swabs, although copy numbers were notably lower than those from nasal samples. Some animals showed fever and appetite loss between 10 and 12 dpi. However, the analysis of the collected organs did not detect viral RNA. Pathological investigation revealed severe perivasculitis and lymphoplasmacytic vasculitis, increased number of type II pneumocytes, macrophages and neutrophils in the alveolar septa and lumen and mild peribronchitis in the lungs of sacrificed ferrets 13 dpi. Ferrets infected through trachea showed viral RNA in the nasal turbinate and soft palate 2, 4, and 8 dpi and in amygdala and trachea only 8 dpi [95]. These results together with the data obtained from the evaluation of infection response in ferrets after inoculation of different viral loads, demonstrated that SARS-CoV-2 can replicate in the upper respiratory tract of these animals, showing a disease pattern similar to humans [96].

In a transmissibility assay, intranasal inoculation of SARS-CoV-2 collected from donor ferrets resulted in respiratory tract infection of recipient animals between 11 and 19 days after inoculation. When transmitted via direct contact between animals, the virus compromised the animals’ respiratory tract 1 to 3 days after exposure. Transmission was also evaluated among independent indirect recipients; animals were disposed in a different compartment separated from infected ferrets by a wall that allowed for air circulation. In these animals, infection was observed between 3 and 7 days. The pattern of viral dissemination in direct and indirect contacts was similar to that observed in inoculated ferrets. Infectious particles could be isolated from all animals, showing that any route was able to infect ferrets efficiently [97]. In another study, naive ferrets kept in direct contact with infected ferrets showed signs of infection such as rise in body temperature between 2- and 6-days post exposure. Ferrets placed in indirect contact with infected animals did not show clinical signs of the disease. However, some indirect contacts were positive for viral RNA, which indicates possible air transmission. Pulmonary histology of ferrets showed only signs of inflammatory process that occurred 4 dpi [98].

Ferrets were also used for assessing the immune response against SARS-CoV-2 after intranasal inoculation. Compared to influenza A infection, SARS-CoV-2 triggered a milder airway immune response. This data was compared to results obtained in humans, using lung samples from post-mortem COVID-19 patients. The analysis showed that genes significantly induced in response to SARS-CoV-2 included a subset of interferon-stimulated gene (ISGs), but not INF-I or IFN-III. SARS-CoV-2 also induced high levels of chemokines, such as CCL2, CCL8, and CCL11. The immune response observed in human samples was similar for ferrets. However, the authors did not report respiratory or systemic pathological signs in the SARS-CoV-2 infected ferrets, suggesting a major limitation in the use of these animals for pathology studies [41].

### 3.4. Dogs

Although dogs present low susceptibility to SARS-CoV-2 infection, these animals can be infected when in contact with COVID-19 human patients. Dogs express ACE2, but there is a single mutation (H34Y) in the canine ACE2 receptor that is not found in human or feline ACE2 and this residue appears to be critical to the low susceptibility of dogs to SARS-CoV-2 [99].

Soon after SARS-CoV-2 emerged in Hong Kong, the Department of Agriculture, Fisheries and Conservation (AFCD) offered the option for pet owners to have their dogs and cats looked after and tested for SARS-CoV-2. Fifteen dogs from households with confirmed COVID-19 cases were tested and two asymptomatic dogs (a 17-years-old neutered male Pomeranian and a 2.5-years-old male German shepherd) were found positive in nasal swabs samples. Immune responses were detected in both dogs using plaque reduction neutralization assays. Molecular analyzes showed that the genome of the viruses isolated from the two dogs were identical to the virus detected in human tutors, suggesting a human-animal transmission of SARS-CoV-2 [100].

Dogs have also been infected with SARS-CoV-2 under experimental conditions. Five three-month-old Beagle dogs were infected with SARS-CoV-2. The virus was inoculated intranasally and the directly infected dogs were allocated together with uninfected animals. Oropharyngeal and rectal swabs were collected 2, 4, 6, 8, 10, 12, and 14 dpi for RNA detection and virus titration. Viral RNA was detected in rectal samples from two dogs 2 dpi and one dog 6 dpi. The dog positive for viral RNA in the rectal sample was sacrificed, but the organs and tissues collected did not have viral RNA. Only two dogs seroconverted, indicating the low susceptibility of these animals to SARS-CoV-2 infection [95]. Although dogs do not develop clinical disease, once inoculated, there is the development of a neutralizing antibody response that begins 14 days after inoculation, peaking at 21 day [101]. A recent study in which dogs were intranasally infected with SARS-CoV-2 showed that the animals seroconverted and mounted a specific neutralizing antibody response, but did not shed the virus upon infection [102].

To date, there is no substantial evidence to support that dogs infected with SARS-CoV-2 can transmit the virus to humans [102], and most dog infections seems to have been acquired from infected humans [103].

### 3.5. Cats

During COVID-19 outbreak in Wuhan, cases of domestic cats infected with SARS-CoV-2 were reported. Cats were diagnosed by serology assays. Infection in cats can occur after contact with COVID-19 patients or due to environmental contamination [104]. Recent studies have shown that domestic cats are susceptible and that infected animals can efficiently transmit SARS-CoV-2 to healthy animals [95,102,105].

Groups of young cats (6–9 months old) and adults (10–14 years old) inoculated via intranasal route had the virus detected in the upper and lower airways. Although the animals did not show signs of the disease, necropsy exams detected interstitial pneumonia, loss of eyelashes and epithelial necrosis, as well as inflammation in the nasal turbinate and trachea. In a group of 10 young cats, two animals died on the third and 13th day after infection. Virus antigen was present in the nasal concha epithelium and necrotic debris was found in the amygdala, submucosal trachea glands, and small intestine enterocytes [95]. Similar findings have been independently obtained from adult animals (5–8 years old) [102].

Cat-to-cat transmission has been reported. Airway samples from healthy cats presented viral RNA after they have been exposed to infected animals [95,102]. There is still no solid evidence proving that cats infected with SARS-CoV-2 are capable of transmitting the virus to humans, although the high levels of viral shedding (10^6^ pfu/mL) in nasal and oropharyngeal swabs for up 10 days is worrisome [102].

### 3.6. Non-Human Primates

Non-human primates (NHP) have been widely used to study SARS-CoV and MERS-CoV infection and are being now explored for COVID-19 research. Studies were performed in cynomolgus macaques (*Macaca fascicularis*), African green monkeys (*Chlorocebus aethiops*), rhesus macaques (*Macaca mulatta*), and in the common marmosets (*Callithrix jacchus*) to better understand the disease pathogenesis, immunity, and testing suitable vaccine and therapeutic approaches [106,107,108,109]. Non-human primates are more phylogenetically similar models to humans than other animal models, and although this is an advantageous feature, it should be noted that studies with them generally use a limited number of individuals (1 or 2 animals per group). Therefore, the results obtained should be interpreted with caution [110].

Cynomolgus monkeys were used as a model to assess the pathogenesis of SARS-CoV-2 and other previous emerging coronaviruses infections. Animals inoculated with SARS-CoV-2 did not show clinical signs of the disease but developed diffuse alveolar damage in types I and II pneumocytes and damage to bronchial and bronchiolar mucosal epithelial cells. In SARS-CoV infection, lung lesions were more severe than those caused by SARS-CoV-2 and milder than those caused by MERS-CoV [106].

Rhesus monkeys infected by the intratracheal route with SARS-CoV-2 developed respiratory diseases between 8 and 16 dpi. High viral loads were detected in nose and throat samples and pulmonary infiltrates were observed after pulmonary radiography. Infected animals also presented viral RNA in samples from rectal swabs. In summary, rhesus monkey manifested a moderate disease similar to the ones observed in most human cases [107]. Ocular conjunctival route has also been evaluated as a form of infection in rhesus monkeys. The model was permissive to SARS-CoV-2 infection leading to mild pneumonia, with no cases of severe pneumonia or death [111].

Lu et al. compared the different susceptibilities of Old World (cynomolgus and rhesus macaques) and New World monkeys (marmosets) to SARS-CoV-2 infection and compared the clinical signs, viral shedding, and replication, and host responses to the infection. All three species were susceptible to SARS-CoV-2 infection as determined by viremia and viral shedding in nasal, pharyngeal, and anal swabs. A viral load was detected in the pulmonary tissues of cynomolgus and rhesus macaques, but not in marmosets. Marmosets were the least susceptible NPH and rhesus monkeys were found to be the most suitable model for COVID-19 as it most closely recapitulated the disease in human patients [108]. At the molecular level, differences in the amino acids sequences of host ACE2 partially explained the differences in susceptibility. *M. mulatta* and *M. fascicularis* have identical amino acids sequences to the human ACE2 region that binds to the RDB domain of the SARS-CoV-2, whereas the *C. jacchus* and humans differ by four amino acids. Mice and ferrets differ from this critical domain of the human ACE2 receptor by eight and seven amino acids, respectively [108].

African green monkeys infected by multiple mucosal routes (nasal, oral, ocular, and tracheal) or through aerosol exposure presented mild clinical infection that caused a transient decrease in pulmonary tidal volume. CT scan revealed lung lesions four days after infection. Infectious virus particles were eliminated by both respiratory and gastrointestinal tracts of all infected animals. Necropsy revealed viral RNA in both respiratory and gastrointestinal systems, with higher levels in the gastrointestinal tract. All infected animals presented anti-SARS-CoV-2 antibodies, and this high frequency of seroconversion has also been observed in human COVID-19 patients [109].

Viral replication in nasopharyngeal, anal, and lung samples is more active in aged monkeys than younger animals. When infected with SARS-CoV-2, monkeys develop typical interstitial pneumonia characterized by thickening of the alveolar septum accompanied by inflammation and edema. Nevertheless, diffuse severe interstitial pneumonia occurs mainly in aged monkeys. This data suggests that models of aged monkeys may be useful to mimic the most severe form of COVID-19 [112].

A crucial question in COVID-19 immunity is whether a previously infected person can be re-infected and develop clinical signs of the disease. NPH models have been used to fill this knowledge gap. Chandrashekar et al. and Deng et al. independently infected rhesus monkeys with SARS-CoV-2 and then re-challenged these animals with the homologous virus. They found that animals indeed developed protective immune responses upon virus re-exposure at either 28 days [113] or 35 days [114] post–initial challenge.

NHP COVID-19 models have paved the way for the development of vaccines [115,116,117,118,119] and antiviral drugs that [120,121] have undergone or are in Phase III clinical trials in several laboratories around the world.

## 4. Other Miscellaneous Models

The susceptibility of other animal hosts to SARS-CoV-2 have been investigated in order to identify animal models and hosts involved in the COVID-19 ecology. Although pigs (*Sus scrofa domesticus*) are susceptible to SARS-CoV infection, several attempts to infect pigs experimentally with SARS-CoV-2 have not been successful [95,122,123], even though some porcine cell lines support productive viral replication. Thus, pigs do not seem to play a role in SARS-CoV-2 dissemination and are not a suitable pre-clinical animal model to study SARS-CoV-2 pathogenesis or efficacy of respective vaccines or therapeutics. Bird species such as chickens and ducks are not susceptible to SARS-CoV-2, as defined by the lack of replication and seroconversion upon virus exposure [95,122].

Fruit bats (*Rousettus aegyptiacus*) experimentally infected with SARS-CoV-2 developed a transient infection, with rhinitis being a histopathological alteration detected without any clinical manifestation. SARS-CoV-2 RNA was detected in the trachea, lung, and lung-associated lymphatic tissue of infected animals and one out of three contact bats became infected, suggesting these animals may serve as a virus reservoir [122].

Raccoon dogs (*Nyctereutes procyonoides*) and minks (*Neovison vison*) are raised for fur production and have been shown to be susceptible to SARS-CoV-2 under experimental and natural conditions, respectively. Minks have been involved in the genesis of a new SARS-CoV-2 variant strain upon acquiring the infection from humans [58]. Infected minks on two farms in the Netherlands showed respiratory disease and increased mortality. In general, animals show signs of watery nasal discharge, although some minks could present severe breathing complications. Interstitial pneumonia was observed and viral RNA was detected in the nasal conchae, lung, throat and rectal swabs. In addition, viral RNA was also detected in the liver of one animal. Different genetic variants of SARS-CoV-2 derived from minks showed to be transmissible between animals from the same farm, but not between minks from other farms [124].

## 5. Concluding Remarks

Few months has passed since the discovery of SARS-CoV-2, nevertheless, great progress has already occurred in the development of in vitro and in vivo models capable of mimicking aspects of viral biology and COVID-19 pathology (Figure 1). These models have been applied in different areas, such as virus characterization and development of vaccines and antiviral therapies. Limitations of each model must be considered during experimental design. The most suitable models will vary according to the question to be answered and should provide reliable and translatable results for humans. 

The use of in vitro models based on monolayer cultures of immortalized cell lines proves to be a faster and more straightforward approach for SARS-CoV-2 studies than primary cells. However, they often do not recapitulate the physiological conditions in vivo and primary cells and explants are an option in these cases. Most animal models developed so far are susceptible to SARS-CoV-2 infection and develop non-fatal diseases of various degrees of severity. Successful development of lethal SARS-CoV-2 models have been achieved by either expressing hACE-2 in mice or by serially passing the virus in conventional mice to produced mouse-adapted SARS-CoV-2.

Each model has its own applicability in studies of the virus and the disease. Mice and hamsters are small animal models that are more easily available and are cheaper for housing, but translation of knowledge from rodent studies to the clinic can be challenging. Ferrets are excellent models for transmission, pathogenesis and countermeasure testing, but their availability in certain countries can be an issue and require special facilities for housing and breeding. Cats are natural hosts for SARS-CoV-2 and transmit the virus well, but do not display typical signs of disease. Non-human primates constitute the model most genetically close to humans, which is important for comparing host responses. However, the development of this model can be expensive and time-consuming. Despite the advances achieved so far, robust in vitro or in vivo models for SARS-CoV-2 infection still need to be determined. Characterization of new models will be important to foster SARS-CoV-2 research and help to control COVID-19 as the pandemic continues to take its toll.

## Figures and Tables

**Figure 1 viruses-13-00379-f001:**
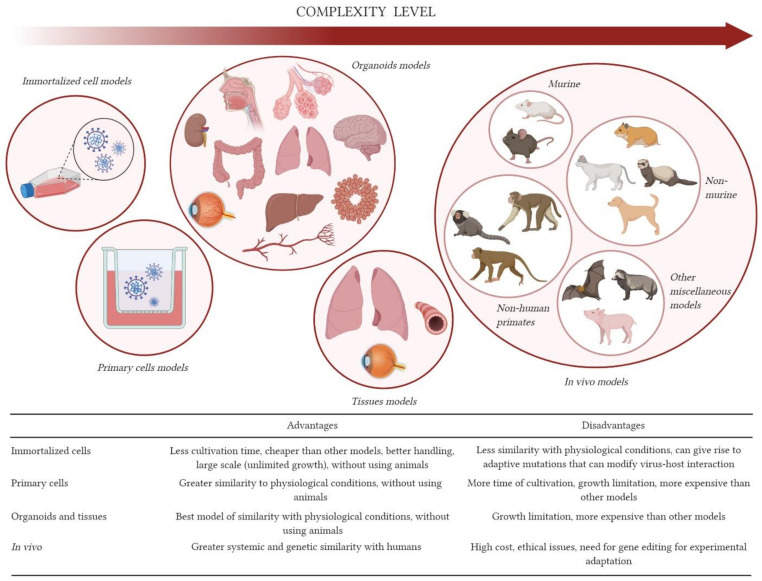
Models for severe acute respiratory syndrome coronavirus 2 (SARS-CoV-2) studies. In vitro, ex vivo and in vivo models previously reported were organized according to complexity level. Advantages and disadvantages of each model are summarized in the table below. Created with BioRender.com.

**Table 1 viruses-13-00379-t001:** In Vitro and ex vivo models for severe acute respiratory syndrome coronavirus 2 (SARS-CoV-2) studies.

Cell/Tissue Type	SARS-CoV-2 Strain	Viral Cultivation	Main Applications	References
Two-dimensional (2D) Models: Immortalized Cells				
Vero CCL-81 and Vero E6	2019-nCoVBetaCoV/Wuhan/WIV04/2019; BetaCoV/Korea/SNU01/2020; SARS-CoV-2/USA-WA1/2020; Australia/VIC01/2020; C-Tan-nCoV Wuhan strain 01	Vero cells used in viral isolation were cultured as monolayers in flasks at 37 °C and 5% atm of carbon dioxide (CO_2_) in Dulbecco’s Modified Eagle Medium (DMEM) or Earle’s Minimum Essential Medium (EMEM) supplemented with 2 to 10% fetal bovine serum (FBS). Some authors cite extra supplementation with antibiotics and buffers. Cytopathic effect could be observed in 2 to 6 days of infection (d.p.i.) or after another viral passage. Protocols for antiviral assays include cultivation in microplates with different values for multiplicity of infection (MOI).	Virus isolation of different SARS-CoV-2 strains; infection characterization; evaluation of host responses; evaluation of antiviral activity; vaccine production.	[15,16,17,18,19,25,26,27,28,29,30]
Huh-7	2019-nCoV BetaCoV/Wuhan/WIV04/2019; SARS-CoV-2/USA-WA1/2020	Huh-7 cells were cultured in DMEM supplemented with 10% FBS at 37 °C and 5% atm CO_2_. Some authors have reported supplementation with antibiotics and antimycotics. For antiviral assays, MOI 0.01 was used, and the supernatant was collected after 48 h.	SARS-CoV-2 isolation; infection characterization; evaluation of host responses; evaluation of antiviral activity.	[15,28]
Caco-2	BetaCoV/Hong Kong/VM20001061/2020	Caco-2 cells were maintained at 37 °C and 5% atm of CO_2_ and cultured with MEM or DMEM supplemented with 10% FBS. Some authors have reported supplementation with antibiotics and antimycotics. SARS-CoV-2 was infected at MOI 0.1, 1 or 2. Detection of viral RNA after 48 h post-infection.	SARS-CoV-2 isolation; infection characterization; evaluation of host responses.	[10,23]
Calu-3	SARS-CoV-2/USA-WA1/2020	Calu-3 cells were cultured in DMEM supplemented with 10 or 20% FBS at 37 °C and 5% atm CO_2_. Some authors have reported supplementation with antibiotics and antimycotics. SARS-CoV-2 was infected at MOI 0.01 or 0.001 and viral presence was evaluated at 24, 48, or 72 h post-infection.	Evaluation of antiviral activity.	[28]
A549-ACE2	SARS-CoV-2/USA-WA1/2020	A549 cells were maintained at 37 °C and 5% atm of CO_2_ and cultured in DMEM supplemented with 10% Serum Plus II. SARS-CoV-2 was infected at MOI 0.1 for 36 h.	Mapping of genes that are associated with SARS-CoV-2 infection.	[24]
Three-dimensional (3D) Models: Explants and Organoids				
Human nasal epithelial cells (HNE), bronchial epithelial cells (large airway respiratory cells—LAE), bronquiolar epithelial cells (small airway respiratory—SAE)	SARS-CoV-2/USA-WA1/2020	HNE cells were grown in an air-liquid system (ALI) system. LAE and SAE cells were initially co-cultured with mitomycin-treated 3T3 J2 cells in DMEM then passed to an ALI system. After cultivation, the viruses were inoculated at MOI 0.5 and 3. Cytopathic effect not described. Viral titers were described 24 h post-infection.	SARS-CoV-2 isolation; infection characterization; evaluation of host responses.	[19,36]
Human alveolar epithelial cells (AECs)	BetaCoV/Hong Kong/VM20001061/2020	Culture of primary cells in flasks. AECs were infected with SARS-CoV-2 at MOI 0.1 or 2. Detection of viral RNA after 24 h.	Infection characterization; evaluation of host responses.	[10]
Normal human bronchial epithelial cells (NHBE)	SARS-CoV-2/USA-WA1/2020	ALI system was not mentioned as a cultivation mechanism. NHBE cells were cultured in bronchial epithelial growth medium supplemented with BEGM SingleQuots. SARS-CoV-2 was infected at MOI 2 for 24 h.	Infection characterization; evaluation of host responses.	[41]
Human tracheobronchial epithelial cells	SARS-CoV-2/USA-WA1/2020	Cultivation of human tranqueobronchial epithelial cells was performed in an ALI system for 6 to 8 weeks. The virus was inoculated at MOI 0.5 and incubated for 48 h.	Infection characterization; evaluation of antiviral activity.	[28,42]
Adult human ocular cells	SARS-CoV-2/USA-WA1/2020	Ocular tissue was maintained in DMEM/F-12 supplemented with FBS and ROCK inhibitor. SARS-CoV-2 was infected in MOI 1 for 24 h. The titrations were performed with Vero E6 cells. Real-time quantitative polymerase chain reaction (qRT-PCR) was also used for analysis.	Infection characterization; evaluation of host responses.	[55]
Human neural progenitor cells (hNPCs)	SARS-CoV-2 HKU-001a; SARSCoV GZ50	hNPCs are derived from induced pluripotent stem cells (iPSCs) grown in mTeSRTM1 medium induced by a cocktail of supplements. After this induction, the cells were cultured in 1:1 DMEM/F-12 and neurobasal medium with extra supplementation. SARS-CoV-2 was infected in a MOI 10 and the infection analyzed by qRT-PCR.	Infection characterization.	[51]
Neurospheres	SARS-CoV-2 HKU-001a; SARSCoV GZ50	Neurospheres are derived from iPSCs and were isolated using Accutase and then maintained in Iscove’s Modified Dulbecco’s Medium (IMDM) supplemented with 15% FBS and additional components. After the formation of the embryoid body (EB), the rosettes were formed in 96-well plates containing DMEM/F12 medium supplemented with fibroblast growth factor 2 (FGF2) and Gem21 NeuroPlex. The formation of neurospheres is given by the rotation of cells without FGF2. A virus title of 2.6 × 10^6^ p.f.u./mL were used to infect the neurospheres. Plates and qRT-PCR assays were performed to detect and quantify SARS-CoV-2.	Infection characterization.	[51]
Brain organoids	SARS-CoV-2 HKU-001a; SARSCoV GZ50	Brain organoids are derived from iPSCs (from donors or not) and were isolated using Accutase and then kept in supplemented mTeSR1 medium for three days, where a series of changes in neurobasal media takes place to form organoids. A 2.6 × 10^6^ p.f.u./mL virus titer or supernatant of SARS-CoV-2 cultivar was used to infect the organoids. qRT-PCR plates and assays were performed to detect and quantify SARS-CoV-2.	Infection characterization.	[50,51]
Liver organoids	SARS-CoV-2/USA-WA1/2020	Liver organoids were derived from human pluripotent stem cells (hPSCs), which were cultured in Matrigel medium and differentiated with Activin A, BMP-4, bFGF and hepatocyte growth factor. SARS-CoV-2 was infected with MOI 0.01, 0.05, and 0.1. The titrations were performed with Vero E6 and HEK293 cells. qRT-PCR was also used for analysis.	Infection characterization.	[47]
Kidney organoids	SARS-CoV-2/human/SWE/01/2020	Kidney organoids were derived from human pluripotent stem cells (hPSCs), which were cultured in RPMI medium with multiple supplements. SARS-CoV-2 was infected at 10^3^ or 10^5^ viral particles and quantified by qRT-PCR.	Infection characterization.	[56]
Lung organoids	SARS-CoV-2/USA-WA1/2020	Lung organoids were derived from human pluripotent stem cells (hPSCs), which were cultured in supplemented DMEM/F12 medium. SARS-CoV-2 was infected at MOI 0.01 for 24 h and quantified by qRT-PCR.	Infection characterization; evaluation of antiviral activity.	[48]
Colonic organoids	SARS-CoV-2/USA-WA1/2020	Colonic organoids were derived from human pluripotent stem cells (hPSCs), which were cultured in supplemented DMEM/F12 or RPMI1640 medium. SARS-CoV-2 was infected at MOI 0.01 for 24 h and quantified by qRT-PCR.	Infection characterization; evaluation of antiviral activity.	[48]
Blood vessels organoids	SARS-CoV-2/human/SWE/01/2020	Blood vessels organoids were derived from human pluripotent stem cells (hPSCs). SARS-CoV-2 was infected at 10^2^, 10^4^ or 10^6^ viral particles and quantified by qRT-PCR.	Infection characterization.	[56]
Eye organoids	SARS-CoV-2/USA-WA1/2020	Ocular organoids were derived from human embryonic stem cells (hESCs), which were cultured in mTeSR1 medium for 10 days and exchanged for SEAM differentiation media to form ocular organoids. SARS-CoV-2 was infected in MOI 1 for 24 h. The titrations were performed with Vero E6 cells. qRT-PCR was also used for analysis.	Infection characterization; evaluation of host responses.	[55]
Bronquial tissue	BetaCoV/Hong Kong/VM20001061/2020	Bronquial tissue were maintained on F12K medium in ALI system. The authors cite extra supplementation with antibiotics and buffers. SARS-CoV-2 was infected for 1 h at 37 °C and washed with PBS buffer after that time. Supernatant samples were collected after 1, 24, 48, 72, and 96 h post-infection. The titrations were performed with Vero E6 or MDCK cells. Histological and immunohistochemical analyzes were also performed.	Infection characterization.	[10]
Lung tissue	BetaCoV/Hong Kong/VM20001061/2020; SARS-CoV-2/human/HKG/HKU-001a/2020	Lung tissue fragments were maintained on supplemented DMEM/F12 medium in plaques or supplemented F12K medium in an ALI system. SARS-CoV-2 was infected for 1 or 2 h at 37 °C and then washed with PBS buffer. Supernatant samples were collected after 1, 2, 24, 48, 72, and 96 h post-infection. Titrations were performed with Vero E6 or MDCK cells. Histological and immunohistochemical analyzes were also performed.	Infection characterization.	[10,46]
Conjunctiva tissue	BetaCoV/Hong Kong/VM20001061/2020	Conjunctiva tissue were maintained on supplemented F12K medium in an ALI system. SARS-CoV-2 was infected for 1 h at 33 °C and then washed with PBS buffer. Supernatant samples were collected after 1, 24, 48, 72, and 96 h post-infection. Titrations were performed with Vero E6 or MDCK cells. Histological and immunohistochemical analyzes were also performed.	Infection characterization.	[10]

d.p.i.: days post infection; MOI: Multiplicity of infection; atm: atmosphere; FBS: Fetal bovine serum; p.f.u.: plaque-forming unit.

**Table 2 viruses-13-00379-t002:** Mouse models used in SARS-CoV-2 studies.

Mouse Strain	Background	Age	Viral Strain	Route of Infection /Dose	Major Findings	Reference
ACE2 (angiotensin-converting enzyme)	C57BL/6	4–5-weeks and 30-weeks-old	BetaCoV/Wuhan/AMMS01/2020	Intranasal/4 × 10^5^ p.f.u./mL or Intragastric/4 × 10^6^ p.f.u./mL	Young and elderly hACE2 (human angiotensin-converting enzyme) 2 mice showed high viral loads in the trachea and brain when inoculated intranasally. Intragastric inoculation led to pathological pulmonary changes.	[65]
HFH4	C3B6	8–10-weeks-old	IVCAS 6.7512	Intranasal × 10^4^ TCID_50_/mL or 7 × 10^5^ TCID_50_/mL	Infected mice had typical interstitial pneumonia. Viral loads were found in the lungs at higher titers, but viral RNA was also found in the eyes, heart and brain. Pre-exposure to SARS-CoV-2 has been shown to protect mice from developing severe pneumonia.	[67]
hACE2	ICR	6–11-months-old	BetaCoV/Wuhan/IVDC-HB-01/2020	Intranasal/1 × 10^5^ TCID_50_/mL	Transgenic hACE2 mice inoculated with SARS-CoV-2 showed interstitial pneumonia. Viral antigens were found in bronchial epithelial cells, alveolar macrophages and alveolar epithelium.	[68]
hACE2	*	*	SARS-CoV-2 MA (clone 14569023)	Intranasal/1 × 10^5^ p.f.u./mL	Through reverse genetics, a recombinant virus was created capable of replicating in upper and lower airways of young and elderly BALB/c mice. The disease was more evident in older animals.	[71]
WT (wild-type)	BALB/c	12-months and 10-weeks-old				
K18-hACE2	C57BL/6	6-weeks-old	USA-WA1/2020	Intranasal/1 × 10^4^ p.f.u./mL	K18-hACE2 mice allow replication of the virus at high titers in the nasal conchae, lung and brain, showing high lethality and production of cytokines and chemokines. Adenovirus-mediated delivery, on the other hand, results in viral replication with lower titers limited to the nasal conchae and lung, with no clinical signs of infection.	[74]
WT	BALB/c	9-months and 6-weeks-old	BetaCov/human/CHN/Beijing_IMEBJ05/2020	Intranasal/7.2 × 10^5^ p.f.u./mL	The MASCp6, an adapted strain of SARS-CoV-2, infected elderly and young mice efficiently, resulting in moderate pneumonia and inflammatory response.	[77]

* Not reported; p.f.u.: plaque formation unit; TCID_50_: Median Tissue Culture Infective Dose; WT: wild-type.

**Table 3 viruses-13-00379-t003:** Non-murine models for SARS-CoV-2 studies (except non-human primates).

Animal Species	Age	Viral Strain	Route of Infection/Dose	Major Findings	Reference
Syrian hamster	6–10-weeks-old	*	Intranasal/10^5^ p.f.u./mL	Animals challenged with SARS-CoV-2 showed viral replication, severe edema, inflammation and cell death in the lungs. The animals also showed weight loss and increased respiratory rate.	[79]
	4–5-weeks-old	BetaCoV/Hong Kong/VM20001061/2020	Intranasal/8 × 10^4^ TCID_50_/mL	Viral antigens were observed in the nasal mucosa, bronchial epithelial cells, duodenal epithelial cells and lung of infected hamsters. Rapid viral clearance and pneumocyte hyperplasia were also found.	[80]
	1-month-old and 6–7-months-old	SARS-CoV-2/UT-NCGM02/Tóquio and SARS-CoV-2/UW-001/Human/2020/Wisconsin	Intranasal and ocular /10^3^, 10^5^, 10^6^ p.f.u./mL	SARS-CoV-2 isolates replicated efficiently in the animals’ lungs, causing severe lung disease. Serious lung injuries were observed. Infected hamsters developed neutralizing antibody responses that prevented infection after viral re-exposure.	[81]
	6–8-weeks and 7–12-weeks-old	BetaCov/Belgium/GHB-03021/2020	Intranasal/2 × 10^5^ TCID_50_/mL or 2 × 10^6^ TCID_50_/mL	Infected wild-type hamsters showed bronchopneumonia and pulmonary inflammatory response with neutrophil infiltration and edema.	[82]
	6-weeks and 32–34-weeks-old	BetaCoV/Germany/BavPat1/2020	Intranasal/1 × 10^5^ p.f.u./mL	The replication of the virus in the upper and lower respiratory tract occurred regardless animals’ age. However, old hamsters had greater weight loss compared to young animals, in addition to developing conspicuous alveolar and perivascular edema. Viral RNA was found in the bronchial epithelium, type I and II alveolar epithelial cells and macrophages.	[83]
	6–10-weeks	*	Intranasal/10^5^ p.f.u./mL	Surgical mask partition for challenged index or naive hamsters significantly reduced the transmission to 25%. Surgical mask partition for challenged index hamsters significantly reduced transmission to only 16.7% of exposed naive hamsters.	[84]
Ferrets	3–4-months-old	F13/environnment/2020/Wuhan and CTan/human/2020Wuhan	Intranasal/1 × 10^5^ p.f.u./mL	Viral RNA was found in nasal washes and rectal swabs from infected ferrets. Fever and appetite loss were observed in some animals. However, RNA was not detected in animals’ organs.	[95]
	7-months-old	Victoria/01/2020	Intranasal/5 × 10^2^/10^4^/10^6^ p.f.u./mL	High and medium viral doses induced a consistent viral infection in the animals’ upper respiratory tract, causing bronchopneumonia (high dose) and bronchointerstitial pneumonia (medium dose).	[96]
	6-months-old	BetaCoV/Munich/BavPat1/2020	Intranasal/6 × 10^5^ TCID_50_/mL	SARS-CoV-2 could be transmitted via direct contact and via air (drops and/or aerosols) between ferrets. Viral RNA was detected in infected animals directly between 1 to 3 days after inoculation and after 7 days in animals infected by indirect contact.	[97]
	12–20-months-old	NMC2019-nCoV02	Intranasal/10^5^ TCID_50_/mL	Infected ferrets exhibited elevated body temperatures and viral replication. The virus was retrieved from nasal samples, saliva, urine, and feces. Viral RNA was detected in the nasal concha, trachea, lungs, and intestine. The study demonstrated the possibility of transmission by direct or indirect contact.	[98]
	4-months-old	USA-WA1/2020	Intranasal/5 × 10^4^ p.f.u./mL	Infected ferrets showed low airway immune responses when compared to Influenza A infection.	[41]
Dogs	3-months-old	CTam-H	Intranasal/1 × 10^5^ p.f.u./mL	Infected dogs had viral RNA detected in rectal samples, but viral RNA was not detected in any other organ or tissue. The study demonstrated a low susceptibility of dogs to SARS-CoV-2.	[95]
	5–6-years-old	WAI/2020WY96	Intranasal/1.4 × 10^5^ p.f.u./mL	Dogs inoculated with SARS-CoV-2 did not develop an evident disease, but the production of neutralizing antibodies after infection was found.	[102]
Cats	70-days-old to 3-months-old	CTam-H	Intranasal/1 × 10^5^ p.f.u./mL	The study showed that cats are susceptible to experimental infection and that virus can be transmitted to uninfected cats when housed together. The virus replicated only in the upper respiratory tract of infected cats, especially in younger animals.	[95]
	6–8-years-old	WAI/2020WY96	Intranasal/3 × 10^5^ p.f.u./mL	The findings showed that cats are highly susceptible to infection by SARS-CoV-2 by maintaining a prolonged period of oral and nasal viral release. It has been reported that infected cats develop neutralizing antibodies that prevent possible reinfection, but there are no clinical signs of the disease. The study also demonstrated the possibility of transition by direct contact between animals.	[102]

* Not reported; p.f.u.: plaque formation unit; TCID_50_: Median Tissue Culture Infective Dose; WT: wild-type.

**Table 4 viruses-13-00379-t004:** Non-human primate models for SARS-CoV-2 studies.

Animal Specie	Age	Viral Strain	Route of Infection/Dose	Major Findings	Reference
Cynomolgus macaques (*Macaca fascicularis*)	4–5-years-old and 15–20-years-old	*	Intratracheal and Intranasal/*	SARS-CoV caused more severe lung injuries than SARS-CoV-2 and milder infection than MERS-CoV in these animals.	[106]
Rhesus macaques (*Macaca mulatta*)	4–6-years-old	nCoV-WA1-2020	Intranasal, Intratracheal and Ocular/4 × 10^5^ TCID_50_/mL	Rhesus monkeys manifest the disease caused by SARS-CoV-2. The animals developed respiratory disease, and high viral loads were found in the nose, throat, and bronchoalveolar lavages.	[107]
Rhesus macaques (*Macaca mulatta*)	3–5-years-old	WH-09/humam/2020/CHN	Ocular Conjuctiva and Intratracheal/1 × 10^6^ TCID_50_/mL	The conjunctival ocular route proved to be efficient for infection of these animals, leading them to develop mild pneumonia. However, the disease did not manifest severely.	[111]
Rhesus macaques (*Macaca mulatta*), Cynomolgus macaques (*Macaca fascicularis*) and Marmoset (*Callithrix jacchus*)	*	*	Intratracheal, Intranasal and Conjuctiva/4.75 × 10^6^ p.f.u./mL and 1 × 10^6^ p.f.u./mL	Two families of monkeys from the Old World and one from the New World were inoculated experimentally with SARS-CoV-2. Among the studied species, *M. mulatta* was the most susceptible to infection followed by *M. fascicularis* and *C. jacchus*.	[108]
Rhesus macaques (*Macaca mulatta*)	3–5-years and 15-years-old	BetaCoV/Wuhan/IVDC-HD-01/2020	Intratracheal/1 × 10^6^ TCID_50_/mL	Viral replication occurred more effectively in elderly monkeys, causing severe interstitial pneumonia. Authors suggest that elderly monkeys are useful to model the severe form of the disease.	[112]
Rhesus macaques (*Macaca mulatta*)	3–5 years-old	*	Intratracheal/1 × 10^6^ TCID_50_/mL	SARS-CoV-2 reinfection was described and its signs were presented. Authors suggest that an initial infection prepares the immune system for a possible new infection.	[113]
Rhesus macaques (*Macaca mulatta*)	6–12-years-old	*	Intratracheal or intranasal/1.1 × 10^4^ p.f.u./mL or 1.1 × 10^5^ p.f.u./mL or 1.1 × 10^6^ p.f.u./mL	SARS-CoV-2 reinfection was described and its signs were presented. Authors suggest that an initial infection prepares the immune and humoral systems for a possible new infection.	[114]
African green monkeys (*Chlorocebus aethiops*)	3–5 years-old	SARS-CoV-2/München-1.1/2020/929 (Munich)	Intranasal, oral, ocular and intratracheal/5 × 10^5^ p.f.u/mL	Infected young monkeys had low fever and the respiratory symptoms were limited to a transient decrease in tidal volume. Viral RNA was found in all airways and gastrointestinal system. All animals seroconverted simultaneously for IgM and IgG.	[109]

* Not reported; p.f.u.: plaque formation unit; TCID_50_: Median Tissue Culture Infective Dose.

## Data Availability

Not applicable.

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
