# Peer review of "In Vitro and In Vivo Models for Studying SARS-CoV-2, the Etiological Agent Responsible for COVID-19 Pandemic"

_viruses, 2021, doi:10.3390/v13030379_

Round 1

Reviewer 1 Report

This is a necessary review of the different models used to study SARS-CoV-2. I particularly like the tables since they provide technical detail that makes this review to stand out from other similar reviews on the topic.

Major issues:

-Regarding the use of immortalized cell lines:

1) the authors should comment on the appearance of adaptive mutations that are not necessarily found in nature and might obscure the interpretation of results (e.g. doi: 10.1099/jgv.0.001481). Also include this issue in the corresponding “disadvantages” in Figure 1.

2) the authors should expand on the conclusions and limitations of the experiments of hydroxychloroquine and ivermectin in Vero cells. This would be of service to the community as those compounds are used clinically by physicians in different parts of the world.

3) the authors should include the use of A549 cells supplemented with ACE2 for the direct comparison of the host response to different viruses (reference #35), and more recently for the characterization of host factors required for SARS-CoV-2 infection and replication using CRISPR screens (doi: 10.1016/j.cell.2020.10.030)

-Regarding the use of organoids: the authors should include in Table 1 (and maybe also in the main text) the usage of lung and colon organoids for the identification of SARS-CoV-2 inhibitors (doi: 10.1038/s41586-020-2901-9)

-Line 233: I am not sure of the proper use of reference #50 in this context. Check instead doi: 10.1038/s41586-020-2787-6

-Regarding Animal models:

1) The authors briefly mention in the concluding remarks that none of the animal models develop fatal diseases. However I believe this is a very important point that should not be undermined and instead should be highlighted and expanded.

2) The authors briefly mention the susceptibility of minks to infection. However given the current enthusiasm as a potential animal model, perhaps the authors could expand on the current knowledge of this particular model. Reference #50 has now been published in Science (doi: 10.1126/science.abe5901). Also include reference doi: 10.2807/1560-7917.ES.2020.25.23.2001005

3) The authors should also include the following work on African green monkeys (doi: 10.1371/journal.ppat.1008903)

Minor issues:

-Line 37: include the following reference: doi: 10.3389/fmed.2020.00370

-Lines 37-38: Thus a considerable number of research models have been developed to mimic the pathophysiology…

-Lines 39, 44 & 54: the same idea is repeated constantly: models that can contribute to the development of antivirals and vaccines.

-Line 108: … the compound showed no cytotoxicity for both cells.

-Line 154-155: … showing an induction of chemokines, cytokines and interleukins, with limited IFN-I and III responses.

-Line 187: Similarly to the work of Qi et al., Singh et al. use scRNA-seq to look at the expression of different factors required for SARS-CoV-2 entry (doi: 10.1016/j.celrep.2020.108175)

-Line 198: Re-write sentence (was?)

-Table 1, Liver organoids, major findings: liter -> other?

Author Response

Reviewer #1:

This is a necessary review of the different models used to study SARS-CoV-2. I particularly like the tables since they provide technical detail that makes this review to stand out from other similar reviews on the topic.

Response: We appreciate the reviewer's comment.

Major issues:

-Regarding the use of immortalized cell lines:

1) the authors should comment on the appearance of adaptive mutations that are not necessarily found in nature and might obscure the interpretation of results (e.g. doi: 10.1099/jgv.0.001481). Also include this issue in the corresponding “disadvantages” in Figure 1.

Response: The suggested comment has been added to the manuscript (lines 83-89 and Figure 1).   

2) the authors should expand on the conclusions and limitations of the experiments of hydroxychloroquine and ivermectin in Vero cells. This would be of service to the community as those compounds are used clinically by physicians in different parts of the world.

Response: The suggested comment has been added to the manuscript (lines 126-136).   

3) the authors should include the use of A549 cells supplemented with ACE2 for the direct comparison of the host response to different viruses (reference #35), and more recently for the characterization of host factors required for SARS-CoV-2 infection and replication using CRISPR screens (doi: 10.1016/j.cell.2020.10.030)

Response: The suggested topic has been added to the manuscript (lines 99-103 and Table 1).   

-Regarding the use of organoids: the authors should include in Table 1 (and maybe also in the main text) the usage of lung and colon organoids for the identification of SARS-CoV-2 inhibitors (doi: 10.1038/s41586-020-2901-9)

Response: The suggested topic has been added to the manuscript (lines 224-227 and Table 1).   

-Line 233: I am not sure of the proper use of reference #50 in this context. Check instead doi: 10.1038/s41586-020-2787-6

Response: The correct reference has been added to the manuscript.   

-Regarding Animal models:

1) The authors briefly mention in the concluding remarks that none of the animal models develop fatal diseases. However I believe this is a very important point that should not be undermined and instead should be highlighted and expanded.

Response: The suggested comment has been added (lines 555-558).

2) The authors briefly mention the susceptibility of minks to infection. However given the current enthusiasm as a potential animal model, perhaps the authors could expand on the current knowledge of this particular model. Reference #50 has now been published in Science (doi: 10.1126/science.abe5901). Also include reference doi: 10.2807/1560-7917.ES.2020.25.23.2001005

Response: The suggested comment has been added to the manuscript and the references were updated (lines 537-543).

3) The authors should also include the following work on African green monkeys (doi: 10.1371/journal.ppat.1008903)

 Response: The suggested topic has been added to the manuscript (lines 498-504 and Table 4).   

Minor issues:

-Line 37: include the following reference: doi: 10.3389/fmed.2020.00370

Response: The suggested reference was added to the manuscript (line 36).   

-Lines 37-38: Thus a considerable number of research models have been developed to mimic the pathophysiology…

Response: All suggested changes have been made in the manuscript (lines 36-37).

-Lines 39, 44 & 54: the same idea is repeated constantly: models that can contribute to the development of antivirals and vaccines.

Response: All suggested changes have been made in the manuscript (the terms were removed from lines 37 and 52)

-Line 108: … the compound showed no cytotoxicity for both cells.

Response: All suggested changes have been made in the manuscript (line 116)

-Line 154-155: … showing an induction of chemokines, cytokines and interleukins, with limited IFN-I and III responses.

Response: All suggested changes have been made in the manuscript (lines 172-173)

-Line 187: Similarly to the work of Qi et al., Singh et al. use scRNA-seq to look at the expression of different factors required for SARS-CoV-2 entry (doi: 10.1016/j.celrep.2020.108175)

Response: The suggested topic and reference have been added to the manuscript (lines 205-207)

-Line 198: Re-write sentence (was?)

Response: All suggested changes have been made in the manuscript (lines 219-221)

-Table 1, Liver organoids, major findings: liter -> other?

Response: All suggested changes have been made in the manuscript (Table 1, Main applications)

Reviewer 2 Report

Review of viruses-1038466

  1. This review compiles reports on cells and animals used to infect SARS-CoV-2. The review is packed with details of each study in an organized manner, although usefulness of some of the details is doubtful. If the review were meant to introduce cells and animals useful for studying SARS-CoV-2, would the detailed description of the experimental results such as lines 99 – 101, for example, have been necessary? However, I think there could be many different styles of reviews and the readers might gain something from such seemingly unnecessary details.
  2. The contents of the tables are largely a repetition of the body text. I recommend simplifying the tables to give a simpler overview of the utilized model system, since anyone who wants to use the same system would eventually go back to the original paper. The sections of “Viral strains” and “Route of infection / dose” look not necessary. “Major findings” might limit to information relevant to the utilized system as an infection model system for general or specific purpose.
  3. There were many blunders. I suggest corrections as follows:

Line 28; anayzes -> analysis

Line 37; number research models -> number of research models

Line 38; developed mimic -> developed to mimic

Line 53; the use animal -> the use of animal

Line 84; HAEs, first abbreviation without full name

Line 108; cytotoxicity -> cytotoxic

Line 134; which as -> which was

Line 154; interleukins mRNA levels -> interleukins in mRNA levels

Line 179; later -> latter

Line 208; which -> what

Line 209; the frequency conjunctivitis -> the low frequency of conjunctivitis

Line 225; that that -> that

Line 234; been -> being

Line 248; was observed that led to -> was observed, which led to

Line 319 -322; the sentences do not make sense. Please clarify.

Line 327; regardless -> regardless of

Line 372; infect efficiently ferrets -> infect ferrets efficiently

Line 408; “the development of a response of antiviral neutralizers“, do you mean “the development of a neutralizing antibody response”?

Line 488; infect experimentally pigs -> infect pigs experimentally

Line 505; capable of mimic aspects -> capable of mimicking aspects

Line 520; transmit well the virus -> transmit the virus well

Author Response

Reviewer #2:

This review compiles reports on cells and animals used to infect SARS-CoV-2. The review is packed with details of each study in an organized manner, although usefulness of some of the details is doubtful. If the review were meant to introduce cells and animals useful for studying SARS-CoV-2, would the detailed description of the experimental results such as lines 99 – 101, for example, have been necessary? However, I think there could be many different styles of reviews and the readers might gain something from such seemingly unnecessary details.

Response: We appreciate the reviewer's comment. The manuscript has been thoroughly revised to attend the suggestions of both reviewers.

The contents of the tables are largely a repetition of the body text. I recommend simplifying the tables to give a simpler overview of the utilized model system, since anyone who wants to use the same system would eventually go back to the original paper. The sections of “Viral strains” and “Route of infection / dose” look not necessary. “Major findings” might limit to information relevant to the utilized system as an infection model system for general or specific purpose.

Response: Changes in “Major findings” section have been made in all tables to attend the reviewer´s suggestion. Regarding the sections “Viral strains” and “Route of infection /dose”, authors feel that these are key information when comparing different cell/animal models. We believe the details in the table will facilitate the comparison of different articles by readers, mainly when they looking for models to study a specific viral strain or route of infection or to identify the methodological differences and limitations of the models described.

There were many blunders. I suggest corrections as follows:

Line 28; anayzes -> analysis

Response: The suggested change has been made in the manuscript (line 28)

Line 37; number research models -> number of research models

Response: The suggested change has been made in the manuscript (line 36)

Line 38; developed mimic -> developed to mimic

Response: The suggested change has been made in the manuscript (line 37)

Line 53; the use animal -> the use of animal

Response: The suggested change has been made in the manuscript (line 51)

Line 84; HAEs, first abbreviation without full name

Response: The suggested change has been made in the manuscript (line 81)

Line 108; cytotoxicity -> cytotoxic

Response: The suggested change has been made in the manuscript (line 116)

Line 134; which as -> which was

Response: The suggested change has been made in the manuscript (line 152)

Line 154; interleukins mRNA levels -> interleukins in mRNA levels

Response: The suggested change has been made in the manuscript (lines 172-173)

Line 179; later -> latter

Response: The suggested change has been made in the manuscript (line 197)

Line 208; which -> what

Response: All suggested changes have been made in the manuscript (what was replaced by which, line 235).

Line 209; the frequency conjunctivitis -> the low frequency of conjunctivitis

Response: The suggested change has been made in the manuscript (line 235)

Line 225; that that -> that

Response: The suggested change has been made in the manuscript (line 252)

Line 234; been -> being

Response: The suggested change has been made in the manuscript (line 260)

Line 248; was observed that led to -> was observed, which led to

Response: The suggested change has been made in the manuscript (line 275)

Line 319 -322; the sentences do not make sense. Please clarify.

Response: The suggested change has been made in the manuscript (lines 343-344)

Line 327; regardless -> regardless of

Response: The suggested change has been made in the manuscript (line 354)

Line 372; infect efficiently ferrets -> infect ferrets efficiently

Response: The suggested change has been made in the manuscript (line 400)

Line 408; “the development of a response of antiviral neutralizers“, do you mean “the development of a neutralizing antibody response”?

Response: Yes, we did. The suggested change has been made in the manuscript (line 436)

Line 488; infect experimentally pigs -> infect pigs experimentally

Response: The suggested change has been made in the manuscript (line 523)

Line 505; capable of mimic aspects -> capable of mimicking aspects

Response: The suggested change has been made in the manuscript (line 546)

Line 520; transmit well the virus -> transmit the virus well

Response: The suggested change has been made in the manuscript (line 564)
